# End-to-Side vs. Free Graft Nerve Reconstruction—Experimental Study on Rats

**DOI:** 10.3390/ijms241310428

**Published:** 2023-06-21

**Authors:** Piotr Czarnecki, Juliusz Huber, Agnieszka Szymankiewicz-Szukała, Michał Górecki, Leszek Romanowski

**Affiliations:** 1Department of Traumatology, Orthopaedics and Hand Surgery, Poznań University of Medical Sciences, 61-545 Poznań, Poland; 2Department of Pathophysiology of Locomotor Organs, Poznań University of Medical Sciences, 61-545 Poznań, Poland

**Keywords:** nerve reconstruction, end-to-side, motor evoked potentials, electroneurography, locomotion analysis, walking track analysis, histomorphometry, rat

## Abstract

The long history of regeneration nerve research indicates many clinical problems with surgical reconstruction to be resolved. One of the promising surgical techniques in specific clinical conditions is end-to-side neurorrhaphy (ETS), described and then repeated with different efficiency in the 1990s of the twentieth century. There are no reliable data on the quality of recipient nerve regeneration, possible donor nerve damage, and epineural window technique necessary to be performed. This research attempts to evaluate the possible regeneration after end-to-side neurorrhaphy, its quality, potential donor nerve damage, and the influence of epineural windows on regeneration efficiency. Forty-five female Wistar rats were divided into three equal groups, and various surgical technics were applied: A—ETS without epineural window, B—ETS with epineural window, and C—free graft reconstruction. The right peroneal nerve was operated on, and the tibial nerve was selected as a donor. After 24 weeks, the regeneration was evaluated by (1) footprint analysis every two weeks with PFI (peroneal nerve function index), TFI (tibial nerve function index), and SFI (sciatic nerve function index) calculations; (2) the amplitude and latency measurements of motor evoked potentials parameters recorded on both sides of the peroneal and tibial nerves when electroneurography with direct sciatic nerve electrical stimulation and indirect magnetic stimulation were applied; (3) histomorphometry with digital conversion of a transverse semithin nerve section, with axon count, fibers diameter, and calculation of axon area with a semiautomated method were performed. There was no statistically significant difference between the groups investigated in all the parameters. The functional indexes stabilized after eight weeks (PFI) and six weeks (TFI and SFI) and were positively time related. The lower amplitude of tibial nerve potential in groups A and B was proven compared to the non-operated side. Neurophysiological parameters of the peroneal nerve did not differ significantly. Histomorphometry revealed significantly lower diameter and area of axons in operated peroneal nerves compared to non-operated nerves. The axon count was at a normal level in every group. Tibial nerve parameters did not differ from non-operated values. Regeneration of the peroneal nerve after ETS was ascertained to be at the same level as in the case of free graft reconstruction. Peroneal nerves after ETS and free graft reconstruction were ascertained to have a lower diameter and area than non-operated ones. The technique of an epineural window does not influence the regeneration result of the peroneal nerve. The tibial nerve motor evoked potentials were characterized by lower amplitudes in ETS groups, which could indicate axonal impairment.

## 1. Introduction

Peripheral nerve damage is associated with significant impairment to the functioning of both upper and lower extremities and therefore has always been a challenge for reconstructive surgeons. The technique, type of nerve reconstruction, and short period between the injury and surgery are important to achieve a satisfactory clinical result [1]. The golden standard of microsurgical nerve reconstruction is an end-to-end epineural nerve suture without significant tension between the nerve stumps. It is impossible to perform such a coaptation in a situation with a significant gap between the stumps [1,2]. One possible solution is a reconstruction using grafts, such as from the sural nerve, while studies indicate a poorer regeneration result compared with a direct epineural suture. In such cases, the greater the gap to be reconstructed, the poorer the result can be obtained [1]. Another solution for small gaps between nerve stumps is an artificial neurotube; nerve allograft; vascularized nerve graft; or a natural substitute, a grafted vein. Using them, it is possible to direct the natural regeneration of nerve fibers, reducing the risk of their dispersion. Unfortunately, they cannot be used for large gaps, and the result of regeneration is best for purely sensory nerves and worse for mixed sensory-motor nerves [3,4,5,6,7,8,9]. Recently, nerve transfers have become increasingly popular in reconstructive surgery, based on additional branches or trunks of nerves of lesser importance for reinnervation of the distal stump of the injured nerve. The intercostal nerves, phrenic nerve, branches of the ulnar, or median nerve are commonly used in this method [10,11]. If the proximal nerve stump is located far from the functionally important muscles that it innervates, then despite the use of the above techniques, the regeneration result will be insignificant or clinically negligible. There are rare situations in which the proximal stump is inaccessible or impossible to find, which results in the inability to perform either a direct suture or a reconstruction of the nerve with free grafts [12,13].

End-to-side nerve repair (ETS) appears to be one of the promising options in the presence of the above-mentioned factors that negatively influence nerve regeneration. The basic concept of nerve repair using this technique is achieving regeneration of the nerve fibers along the distal stump of the cut nerve by inducing lateral axon sprouting from an adjacent intact nerve. The efficacy of the ETS is controversial and is still being evaluated clinically [13,14,15,16]. Although there are reports of good nerve regeneration [12,13,15,17], some authors have not shown signs of motor or sensory reinnervation in their reconstruction cases or considered it a poor result [18,19]. In addition, other concerns are raised regarding the possible denervation of the target area supplied by the donor nerve and changes in the connections in the central nervous system [15].

The efficacy of ETS has not been clearly presented, based on an analysis of the literature and studies carried out to date. Available publications that describe different methodologies are often based on limited material, and the resulting statistical confidence is low. Most often, the authors compare the end-to-side suture with the end-to-end suture, which in our opinion is inadequate, because in clinical situations where end-to-end suture is possible (which is the gold standard in nerve coaptation), end-to-side suture is not considered. It is much more reasonable to compare the methods that can be clinically considered in the case of the impossibility of making end-to-end suture, as in the situations with a nerve gap. That was the reason for designing the study to compare end-to-side with graft reconstruction (the gold standard for bridging nerve gaps). Moreover, it is also unclear whether opening the epineural sheath of the donor’s nerve affects the quality of regeneration of the recipient’s nerve and whether it causes any dysfunction of the donor’s nerve.

This study aimed to investigate with the locomotion, morphometric, and electrophysiological methods the efficacy of the common peroneal nerve regeneration in rats after its reconstruction using an end-to-side suture and to compare the results of its regeneration with those in the classical technique of reconstruction with a free graft. We also evaluated the influence of creating an epineural window in the donor nerve on the recipient nerve’s regeneration and the possible impairment of the donor nerve function.

We hypothesize that ETS can be effective but inferior to nerve grafting, and the epineural window procedure can positively influence the final results.

## 2. Results

### 2.1. Results of the Analysis the Walking Track

After the surgery, the values of the indexes gradually increased in each of the studied groups. TFI and SFI values stabilized in week six and PFI in week eight. From that point on, the values of the indexes remained at a similar level until the end of the follow-up period, i.e., over 24 weeks (ANOVA test). At the endpoint, there were no statistically significant differences in the individual indexes between the study groups (ANOVA, post-hoc LSD tests).

The indexes (PFI, TFI, SFI) and the observation time demonstrated a statistically significant positive correlation. A statistically significant positive correlation was determined between the PFI and TFI values in the studied groups at the end point of observation. This indicated a significant influence of the follow-up time on the values of the indexes, which increased to week six (TFI and SFI) and week eight (SFI), and an improvement in the function of the peroneal nerve together with an improvement in the function of the tibial nerve. The highest correlation coefficient was achieved in the end-to-side nerve reconstruction group with the formation of an epineural window and the lowest in the reconstruction group using a free nerve graft (Table 1).

### 2.2. Results of the Electroneurographic Evaluation

Using Student’s t-test, no statistically significant differences were obtained between the amplitude and latency values of the peroneal nerve on the operated side versus the non-operated side in direct and indirect MEP stimulation. However, a statistically significant decrease in the amplitude value on the operated side compared with the non-operated side was obtained for the tibial nerve in direct stimulation in groups A and B and in indirect stimulation of MEP in group B. No statistically significant differences were found in the latency values.

Using the ANOVA analysis of variance, no significant statistical differences were found in the amplitude or latency values between the study groups, neither for the peroneal nor tibial nerve (Table 2).

### 2.3. Histomorphometric Examination Evaluation

Using the ANOVA variance analysis with the Scheffe post-hoc test, we compared differences in morphometric parameters between groups and the norm for the peroneal and tibial nerves.

Histological evaluation of the operated peroneal nerves revealed no significant difference in the number of myelinated nerve fibers compared to the same nerve on the non-operated side. Still, their diameter and surface area were significantly smaller. There was no significant difference in the values of the histomorphometric parameters between different studied groups—all of them achieved the same level of morphological regeneration.

Histologically, the tibial nerve in the studied group, which served as the donor nerve in groups A and B, did not differ significantly from the nerve in the control group. There was no significant difference in the assessed parameters between the group in which the tibial nerve epineurium was open or not (no significant differences between groups A and B) (Figure 1, Figure 2, Figure 3 and Figure 4).

A significant positive correlation was obtained between the diameter of axons of the tibial and peroneal nerves on the operated side in group B. No relation was detected in any group’s number and diameter of axons. The absence of negative correlations indicates that higher values of the peroneal nerve parameters were not associated with their decrease in the tibial (donor) nerve (Table 3).

## 3. Discussion

### 3.1. Analysis of the Walking Track

The walking track analysis of rats was introduced in the 1980s and has evolved [20,21,22,23,24]. It is now a generally accepted method of assessing rats’ sciatic, peroneal, and tibial nerve function recovery. The advantages of this method are the ability to repeatedly evaluate and thus track regeneration, non-invasiveness (no need for anesthesia or surgery), comparability between researchers, simplicity of execution, and no need for expensive equipment. The disadvantages of this method are the difficult interpretation of the results due to the large adaptive capabilities of the rat and the difficulty in obtaining a correct paw print in a situation of large motor inabilities or secondary contractures.

To reduce measurement error and improve the input and conversion of the results, computerized image analysis was used. With this technique, one can perform calculations anywhere, easily archive the material, and view individual results. High magnification in determining the appropriate trace parameters allowed us to reduce the error from measuring the value on paper with a ruler. Magnification allowed us to evaluate even weak prints, often obtained at the early stages of the function recovery when the animal does not use its extremity efficiently.

Another factor in assessing the paw print is the time of observation. In our experiment, all values positively correlated with the follow-up time. The main publications indicate a period of 6–8 weeks as the time needed to stabilize the values of evaluated indexes (PFI, TFI, SFI) [20,25,26]. Our results are consistent with these observations. In the assessment of the peroneal nerve, the time was longer (eight weeks), and for the other nerves, it was shorter (six weeks). Moreover, the PFI values achieved at the end were lower than those for other nerves (TFI, SFI). This is a natural phenomenon associated with the concept of the experiment—damage to the peroneal nerve. The values of the parameters normalize relatively early, although we believe it is important to evaluate them at a later stage. Many authors provide a shorter follow-up time without paying attention to the possible late appearance of secondary contractures and deformations of the paw that may change the values of the indexes [26,27]. In our study, we extended the follow-up period to 24 weeks (about one-fifth of the animal’s life), which allowed us to assess the functions after complete regeneration. We did not notice any significant change in the values of the indexes within that time.

The presented results revealed no differences in the functional assessment of the nerves of the walking track analysis at the endpoint, which indicated the same result of regeneration of the peroneal nerve after an end-to-side suture and a free graft. It should be emphasized that the animal did not achieve normal values. Still, it can be concluded that these are good results (on a scale of 0—normal function, −100—lack of function), both for the peroneal nerve (mean −15.79, SD 17.40) and tibial nerve (mean −9.96, SD 14.43), and in the total assessment of the sciatic nerve (mean −10.99, SD 12.91). These end-to-side (with epineural window) group results are similar to those reported in the literature by Kerns et al. [26]. They assessed only one control group, which had end-to-side nerve grafting of the tibial nerve to bridge a neuroma-in-continuity on rats. The walking-track analysis in both control and experimental groups was performed. Seven days after the lesion was administered, the mean functional loss for all animals was similar. After 21 days of recovery, there was highly significant (*p* < 0.001) improvement over time for both groups, but there were no significant group differences at any time point (*p* > 0.05), as in our group. Similar to our results, at the end of the cited experiment (7 weeks post-repair), neither group was back to the preoperative normal value (*p* < 0.01).

In addition, we demonstrated a positive correlation between PFI and TFI, which was stronger in groups A and B (end-to-side suture). This may indicate a mutual influence of donor nerve parameters on the values achieved in the recipient’s nerve and additionally excludes impairment of donor nerve function associated with collateral regeneration; no significant negative correlations were found. The highest Pearson correlation coefficient values (0.84) were found in group B, which may suggest the result of the epineurium opening on better nerve regeneration in the functional assessment. However, this was not confirmed in other tests discussed below (electroneurophysiology and histomorphometry). The walking track analysis may not be the only test on which the assessment of nerve regeneration is based. Still, it allows the dynamics to be tracked and—thanks to the standardization of the method—comparisons of the results with those of other studies [21,25,28].

### 3.2. Neurophysiological Evaluation

This test used a widely described classical technique, including stimulation and recording potentials with bipolar electrodes [29]. The authors describing this test point to the high susceptibility of the method to interference, the technique of execution, and the sensitivity of the examined nerve to the conditions of the examination [30]. Especially important is the way the electrodes are applied; the absence of nerve tension during the test; the proper placement of the grounding electrode; and the temperature, drying, and appropriate isolation from surrounding tissue. In this study, an attempt was made to exclude the influence of these factors by checking the position of the grounding electrode, isolation, and the absence of nerve tension at the site of contact of the recording and stimulating electrode and moistening the tissues with physiological salt. No recording was obtained several times, including from a healthy nerve, eliminating some animals from the evaluation. This outcome is difficult to explain and could result from equipment malfunction. In most cases, the potentials recorded by the electrical (direct) stimulation method had low excitation thresholds (1 mA). This may indicate that the nerve fibers in the reconstructed nerves were conducting properly. The results of 24 weeks of observation after the surgery suggest the completeness of regeneration. The amplitude and latency parameters obtained for the peroneal nerve did not differ significantly in the individual groups and did not fluctuate from the normal values. Some authors report poorer electrophysiological parameters of the reconstructed nerve [31,32,33]. Other researchers report a complete normalization of values after a follow-up period (up to nine months) [34].

In the literature, the electrophysiological parameters of the donor’s nerve do not differ from those of a non-operated nerve [34]. In groups with donor’s nerve (groups A and B), we observed significant neurophysiological deviations from the norm in the tibial nerve, distorting the picture of the absence of changes in the donor’s nerve histomorphometric evaluation. It may suggest possible damage or impairment to the function of this nerve during the procedure of suturing the peroneal nerve, or they may be associated with a decrease in parameters in favor of the regenerating peroneal nerve, which affects the conduction parameters.

Some authors found a steady increase in the electroneurophysiological values during the follow-up period until about 40 weeks after the surgery in cyclic studies [30,32]. This is interesting, as the parameters of function evaluation (PFI, SFI, TFI) stabilize much earlier. It might indicate a continuous regeneration process with a tendency to improve. An additional new element not described in the available literature is oververtebral stimulation at the lumbar level with a magnetic coil. There are scarce descriptions of similar techniques [35,36,37,38]. We used the smallest available coil, with the parameters of the stimulating beam diameter <5 nm and penetrating the tissues from the point of surface stimulation at 3 cm and a stimulus strength of 70%. It would probably have been better to use a smaller coil dedicated to small animals to eliminate possible interference due to the proximity of the recording electrode and to direct the stimulation precisely.

Examination of motor evoked potentials induced by magnetic fields was a valuable complement to the classically performed electrical stimulation. These studies prove a significant correlation between the results obtained using both stimulation methods [38]. Examination of this type can eliminate such problems as the close location of the stimulating electrode, additional stretching of the nerve at the site of its application, and disturbances resulting from direct muscle stimulation due to tissue conduction. However, it is difficult to conduct this examination correctly due to the method’s sensitivity to the coil’s orientation, its distance from the spinal cord, and especially the movement artifacts of the animal during stimulation (many muscle groups are stimulated simultaneously). The method requires further development and refinement, but it seems promising. In the literature, an additional evaluation of the conduction rate is presented [30,34]. We did not analyze the parameter of the conduction velocity in our study. Its accuracy depends on the conduction distance between stimulation and recording points, which was always constant but too short to provide the reliable calculation of the conduction velocities values taking into account the latencies’ measurement. The methodological weakness for recordings of nerve conduction velocities in small animals during ENG studies was widely discussed in previous experimental works [26,27].

### 3.3. Histomorphometric Evaluation

The applied light microscopy examination provided very good material for a morphometric evaluation, a valuable data source for statistical analysis. Selected images often described in the literature present regeneration but do not allow for assessing its quality and degree compared to a normal nerve. For correct morphometric analysis, it is necessary to have a preparation with adequate quality, equipment for digital conversion, and a technique used for particle counting [39]. At each stage, problems and disruptions affecting the result can occur. Poor image resolution, uneven field illumination, an inaccurate cross-section through the nerve, and incorrectly set particle counting parameters all cause significant discrepancies in the results.

Various parameters of morphometric evaluation of the nerve have been proposed: the number of axons, the degree of their myelination (thickness of myelin to the diameter of the fiber), surface area, diameter, and derived values such as density (amount per unit of the surface area). This study’s simplest and most important parameters were chosen and used, as often described in the literature. The available methods included manual, semi-automatic, and automatic methods. Authors often describe the manual method as time-consuming and the automatic method as too inaccurate [40,41,42]. In our study, we used a semi-automatic method, which influenced the subsequent stages of assessment, especially in the preparation of the image for calculations, where they were modified according to the nature of the image [39,40,43].

The main problems encountered are misinterpretations (over- and under-estimation) described by other authors, taking blood vessels or connective tissue elements as nerve fibers into account. The strict setting of the roundness parameter and the minimum particle size for counting also did not eliminate all longitudinally cut fibers. It was mostly noticed in the regenerating operated peroneal nerve, due to the nature of nerve regeneration than can be partially chaotic. These errors were mostly eliminated by better preparation of the digital material—proper graphics processing, removal of elements that caused errors, measuring more fields, and averaging the result. Even though the SDs for these measurements are quite high, which is the limitation of the study, the authors are aware of it, but still valuable statistical analysis could have been obtained to compare the results from the groups analyzed.

The obtained results confirm the regeneration of the nerve sutured end-to-side. The histological quality of such a nerve is lower than that of a healthy nerve, but it does not differ from a nerve reconstructed with a free graft. It is worth noting that the regeneration process is active, and the number of fibers does not differ from the one in a healthy nerve. Still, they are smaller, as a large part of the nerve is filled with a scar of connective tissue. Even with a small size and number of nerve fibers, the clinical effect may be sufficient for a good functioning [44]. The presented results of the groups with end-to-side neurorrhaphy are similar to those described in the literature by Kerns et al. and Zhang et al. [26,45]. At the same time, some authors report better histomorphometric parameters in nerves reconstructed with an opened epineurium of the donor nerve [12,46]. In this study, there was no difference in the values of the assessed parameters between the group with open (B) and intact (A) epineurium of the donor’s nerve.

In histological assessment, the tibial nerve as a donor on the operated side did not differ significantly from a healthy one. However, it is not consistent with the results of the electroneurophysiological examination. These results may be caused by damage to the sheath of the donor’s nerve (suture, epineural window), which affects the conduction parameters. Additionally, the absence of a negative correlation between the values obtained for both nerves (donor and recipient) may indicate the absence of a negative result of collateral regeneration on the histological properties of the donor’s nerve. In future studies, measuring the thickness of the myelin sheath would provide additional information on the quality and effectiveness of the applied grafts. The additional clinical evidence (walking track analysis) suggests no impairment of this nerve’s function.

### 3.4. Summary

Our tests have shown no difference between the end-to-side suture and the reconstruction with a free nerve graft. No difference was observed between the technique involving and not involving the opening of the epineurium. The donor nerve in most tests did not differ from a non-operated one, which may indicate that its function is not impaired. The only difference was a significantly lower amplitude of the electrical response from the tibial nerve, which may indicate axonal damage.

We are aware about outcomes of rat trails, the ability to regenerate, and the lack of indirect translation of these results into human neuroregeneration. Still, in a clinical setup, there are difficult situations to solve—the posttraumatic nerve gap that is too big to be reconstructed with the grafts with a predictable or any result, as well as the missing proximal stump such as in brachial plexus root avulsion. In some of these situations, development of nerve transfers is the best solution that could be offered to the patient, but it is not always possible to apply without the risk of loss in important function. Moreover, there is also a big development in the last years concerning reverse end-to-side techniques supporting distal muscles for the time of regeneration of the repaired nerve. These protective techniques called “babysitting” or “supercharge” are also based on such experimental trials supporting the idea of end-to-side phenomenon.

In summary, our study provides further support for using the end-to-side technique in salvage situations of nerve gap reconstruction, giving the surgeon another possible weapon in rare but difficult clinical situations.

## 4. Materials and Methods

### 4.1. Animals

The study followed the guidelines in the Helsinki Declaration and was approved by the Bioethics Committee at the Poznan University of Medical Sciences (decision no. 1279/18). The study involved 45 Wistar female rats weighing from 197 to 260 g (mean 229 g, SD ± 14 g). The animals were randomly divided into three groups depending on the type of experimental procedures performed (Figure 5):A.end-to-side suture without an incision of the donor nerve epineurium;B.end-to-side suture with an incision of the donor nerve epineurium;C.reconstruction with a free nerve graft.

The donor in group A and B was the tibial nerve, and the recipient was the reconstructed peroneal nerve. This choice was based on the twice larger diameter of the tibial nerve, which technically makes it possible to suture the peroneal nerve to it. The tibial nerve also has almost twice the number of nerve fibers, which may represent a greater potential for regeneration of the donor’s nerve and may correspond to the clinical situation.

### 4.2. Anesthesia and Surgery

The animals were anesthetized intraperitoneally with ketamine (Ketanest, Park Davis, Vechta, Germany) at a dose of 90 mg/kg of the body mass. The anesthesia started working about 20 min after administration, and its effect lasted about 2 h. In addition, the operated area was anesthetized locally with a 1% lignocaine solution. The depth of anesthesia was verified by a pupil dilatation, and no toe pinch reflex could be elicited after about 25 min. The attempts to measure the blood pressure and PCO_2_ with a small cannula from the external carotid vein were abandoned because they may influence the arterial flow and the neurophysiological parameters. To prevent aspiration of saliva into the lungs, the animals were premedicated with atropine sulfate (Polpharma, Warsaw, Poland) applied by the intramuscular injections at a dose of 0.05 mg/kg [47].

Surgeries were performed in aseptic conditions, using appropriate microsurgical instruments and magnification by a surgical microscope. The operated side was the right side, and the opposing side served as the control. The skin was incised along the edge of the iliac crest. Then, layers of the gluteal muscles were exposed, reaching the space where the sciatic nerve divides into the peroneal, tibial, and sural nerves. After the excision of the peroneal nerve approximately 2 cm from its branching site, the following procedures were performed (Figure 5):A.group A—suturing the stump of the distal peroneal nerve to the side of the tibial nerve;B.group B—as above, but with a previous creation of an oval window in the epineurium of the tibial nerve;C.group C—excision of a fragment of the peroneal nerve with a length of about 1 cm, then suturing it to the damaged nerve after inverting it by 180° as a free graft.

The window was made by lifting the epineurium with micro forceps and then incising it with micro scissors to a circumference matching to the diameter of the sural nerve [48].

Using a locally harvested peroneal nerve fragment eliminates another donor site morbidity that could possibly affect the function and the assessment. Rotating the graft by 180° increases the reliability of the reconstruction without affecting its results, and it was described as a model of graft reconstruction in experimental studies [49].

The proximal part of the peroneal nerve was resected to the point where it branched from the sciatic nerve to prevent possible spontaneous regeneration. A minimum number of Ethilon 10-0 (Ethicon) non-absorbable monofilament sutures were used to ensure proper orientation and strength of the fusion: three sutures in groups A and B and two sutures in group C per repair.

After awakening and general health control, the animals were transferred to the animal room, receiving food and water as required, wherein they remained until the end of the study for 24 weeks. At the end of the study, after walking track analysis, electrophysiological evaluation, and taking specimens for histomorphology, the animals were euthanized using ketamine at twice the lethal dose.

### 4.3. Walking Track Analysis

The analysis is based on the determination of the print length (PL), the distance between toes 1 and 5 (TS—toe spread) and 2 and 4 (ITS—intermediate toe spread), and then on the values of the experimentally obtained parameters. These parameters serve to evaluate the function of the sciatic nerve (SFI—sciatic function index), peroneal nerve (PFI—peroneal function index), and tibial nerve (TFI—tibial function index). The indexes are calculated according to formulas determined experimentally and confirmed by statistical analyses based on previous publications [19,20]. The resulting values ranged from −100—no function to 0—full function.

The indexes were calculated using the following formulas:PFI = 174.9 (EPL-NPL)/NPL + 80.3 (ETS-NTS)/NTS-13.4.TFI = −37.2 (EPL-NPL)/NPL + 104.4 (ETS-NTS)/NTS + 45.6 (EITS-NITS)/NITS-8.8.SFI = −38.8 (EPL-NPL)/NPL + 109.5 (ETS-NTS)/NTS + 13.3 (EITS-NITS)/NITS-8.8.

The designations E (EPL, ETS, EITS) were applied to the corresponding parameters on the operated (experimental) side and N (NPL, NTS, NITS) to the non-operated side (Figure 6A).

Tests were conducted every two weeks, with the first one two weeks after the surgery. The walking track analysis of the rats was dipped in red office ink for the right paw and in blue ink for the left paw (Figure 6B). The animal was placed on a specially prepared paper-lined path, marked with the test date and animal number. The prints were scanned using a Hewlett-Packard flat scanner at 300 dpi and saved in a JPEG format. The image was analyzed using the ImageJ program (National Institute of Health), where measurements were conducted at an appropriate magnification. The resulting values were calculated using Microsoft Excel.

### 4.4. Electrophysiological Evaluation

The examination of nerve impulse conduction was intended to determine the regeneration of the peroneal nerve (recipient) and possible damage to the tibial nerve (donor). The control group provided electroneurographic data for the corresponding nerve trunks on the non-operated side, partially eliminating individual variation in the measured values. Tests were performed 24 weeks after the surgery using the surgical access described earlier.

Two stimulation methods were applied as follows [47]:Motor fibers of the sciatic nerve were electrically stimulated by a pair of silver bipolar hook-shaped electrodes applied to the nerve 1 cm from the fusion site. The cathode was distal, while the anode of the electrode pair was oriented closer to the spinal center. Such an orientation of the simulating poles guarantees the orthodromic excitation of the motor fibers within the nerve. The distance between the recording electrodes poles was 3–4 mm. The ground electrode was placed near the recording electrodes on the muscle. During electroneurographic ENG nerve-to-nerve recordings, special attention was paid not to dry the dissected nerve branches; they were soaked with drops of warm paraffin oil. Electroneurography of the sciatic nerves was applied for bilateral detection of changes in the conduction of neural impulses following surgical nerve grafts. The ENG potentials were recorded from the distal parts of peroneal and tibial nerves with pairs of bipolar silver electrodes, after the application of electrical, rectangular pulses with 0.2 ms duration, at 1 Hz, and intensity from 0 to 20 mA delivered from the other bipolar stimulating silver electrodes in the proximal part of the sciatic nerve. A distance of 3 mm between the recording anode and cathode was maintained. Potentials were recorded in order to verify the conduction of neuronal impulses in the peripheral motor fibers. The recordings were performed at an amplification of 5–5000 µV and a time base of 2–10 ms. The parameters of amplitudes (in µV) and latencies (in ms) in recorded potentials were the outcome measures. Donor tibial and graft peroneal nerve fibers were excited following electrical stimulations of sciatic nerve rectangular pulses with a duration of 0.1 ms at 1 Hz and strength from 0.06 to 1 mA delivered from the KeyPoint device stimulator.Descending fibers of the white matter of the spinal cord were stimulated by a stream of the magnetic field released from the electromagnetic coil over the spine and recorded using the MEP technique (motor evoked potentials induced by a magnetic field). They aimed to evaluate the total efferent neural conduction from lumbar spinal centers to the distal parts of nerves and effectors. Standard single pulses of the magnetic field were used for oververtebral stimulation to induce motor-evoked potentials (MEPs). They were induced from the MagPro R30 (Medtronic A/S, Skovlunde, Denmark) using a 50 mm diameter circular coil placed bilaterally over the descending fibers of white matter at the L3–L5 spinal cord level, and recordings were performed using the MEP technique 10 mm from the peripheral graft. The optimal site for stimulation was defined with tracking stimuli delivered at 1 Hz from 5–60% of maximal output stimulus strength (1.5 T—Tesla), while the maximal amplitude of MEPs was recorded from the nerve. The maximal stimulus output at less than 60% was the highest intensity stimulus applied. The recordings were performed at an amplification of 50–5000 µV and a time base of 2–5 ms. All MEPs recordings were made at 0.5 Hz low-pass filter settings, while the upper-pass filter of KeyPoint was set at 2 kHz. The outcome measures were the parameters of amplitudes (in µV) and latencies (in ms) of MEPs.

The recorded sites were the peroneal and tibial nerves (Figure 7). The tests were carried out using a KeyPoint device (direct stimulation of the nerve) and a MagPro device (spinal stimulation with an electromagnetic coil) manufactured by Medtronic, in the same room, at 21–23 °C. Minimally invasive magnetic and electrical stimuli were used. The strength of the stimuli was adjusted guaranteeing the smallest movement artifact of the stimulated object that could affect the recording conditions while eliciting MEPs and ENG potentials with supramaximal amplitude.

### 4.5. Histomorphometric Evaluation

A morphometric evaluation was applied to assess the quantitative elements of the nerve objectively. A semi-automatic method was used, which influenced the subsequent stages of assessment, especially in the preparation of the image for calculations, where they were modified according to the nature of the image. The material was obtained immediately after the electrophysiological examination. Preparations of the peroneal and tibial nerves were evaluated on the operated side in all animals and on the non-operated side in 15 randomly selected animals (standard). The test material was taken 1 cm from the fusion site and then transferred to a laboratory. Slides for histological evaluation were prepared in the electron microscopy process and modified for light microscopy as follows, and tissues were placed in Karnowski’s solution at 4 °C and pH 7.34. Then, they were placed in a buffer, then dehydrated in increasing concentrations of ethanol. Following dehydration, specimens were embedded in epoxy resin.

The next stages of preparation of the slides included staining with toluidine blue, and making semi-thin sections 0.5 µm, which were placed on a microscope slide [35]. The preparations were then analyzed using a light microscope at 40× lens magnification in the Morphometry and Medical Image Processing Laboratory. Digital recordings of three fields (from each preparation) were made using a camera and the Motic system. For morphometric analysis, ImageJ software version 1.3 with suitable plugins was used. The preparation process included scaling the measurement units, noise reduction, conversion to an 8-bit grayscale image, thresholding to expose the nerve fibers, and then inversion (Figure 8).

As a result of the morphometric analysis (counting of particles) with constant parameters in size range of the detected particles, as well as the degree of their roundness (with the elimination of diagonally cut fibers), the following values were determined in three separate fields of the preparation and averaged:number of nerve fibers in the field;diameter of nerve fibers without myelin sheath;nerve fiber surface area without myelin sheath.

Since the counted particles were not round, the Feret diameter was determined, i.e., the distance between the lines parallel to the edges of the particle.

### 4.6. Statistical Analysis

The results were analyzed using Statistica 13.1 (Statsoft) software. The choice of methods and tests depended on the scale on which the analyzed variables were described and the nature of the distribution of the results and possible relationships between them. Quantitative characteristics were described with basic statistics, including minimum and maximum values (range) and mean and standard deviation (SD) for measurable values. The existence assessed any significant differences between variables in individual groups and in the same group but at different points in the study and the possible existence of correlations between variables. In each case, we used the appropriate post-hoc tests such as the LSD or Scheffe post-hoc tests. *p*-values less than 0.05 were statistically significant.

## 5. Conclusions

This study compared techniques applicable in similar clinical conditions—treating nerve injury with nerve gap. This was novel, contrary to other research comparing end-to-side sutures to end-to-end sutures used in completely different clinical situations. Regeneration of the peroneal nerve after coaptation to the side of the tibial nerve in the rat occurs and is as effective as the reconstruction with a free nerve graft. The diameter and surface area of the nerve fibers of the peroneal nerve after an end-to-side coaptation and reconstruction with a free graft are significantly smaller than those in a non-operated nerve. The opening of the epineurium in the tibial nerve does not significantly affect the regeneration of the peroneal nerve. The tibial (donor) nerve significantly reduced the conduction parameters (amplitude) in neurophysiological tests compared to a non-operated nerve, which may indicate axonal damage. This damage does not cause clinical impairment in the rat’s lower extremity function.

## Figures and Tables

**Figure 1 ijms-24-10428-f001:**
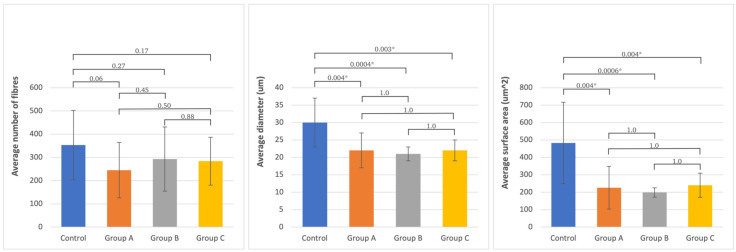
The average number and standard deviation of fibers, diameter, and surface area of axons in the studied groups and in the non-operated **peroneal nerve**. Statistical significance between each group is presented, where * is *p* < 0.05.

**Figure 2 ijms-24-10428-f002:**
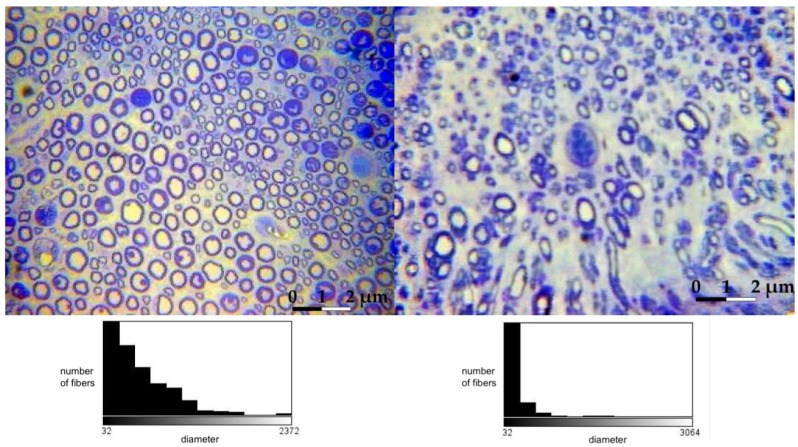
Microphotographs of transverse sections of the **peroneal nerve;** 40× image magnification (left side—non-operated, right side—operated: example from group A). Histograms presenting the fiber count are presented below each of the photographs.

**Figure 3 ijms-24-10428-f003:**
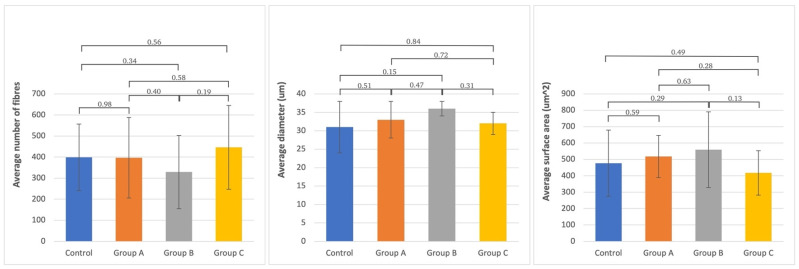
The average number and standard deviation of fibers, diameter, and surface area of axons in the studied groups and in the non-operated **tibial nerve**. Statistical significance between each group is presented, where * is *p* < 0.05.

**Figure 4 ijms-24-10428-f004:**
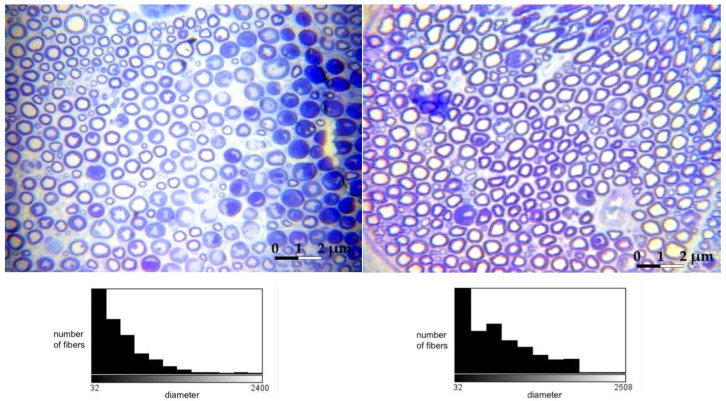
Microphotographs of transverse sections of the **tibial nerve;** 40× image magnification (left side—non-operated, right side—operated: example from group A). Histograms presenting the fiber count are presented below each of the photographs.

**Figure 5 ijms-24-10428-f005:**
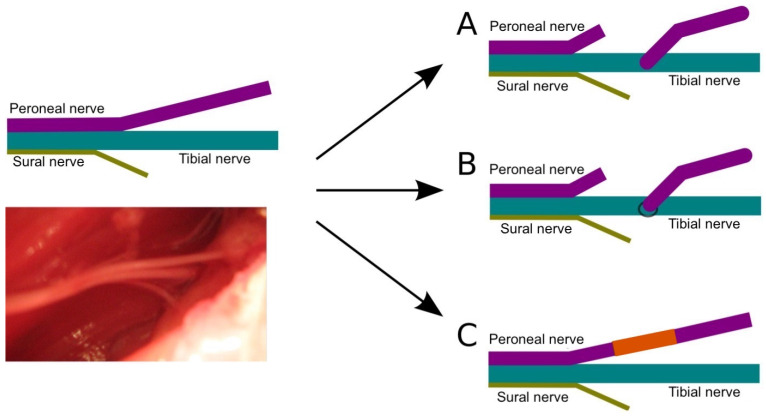
The scheme presents the anatomical division of the sciatic nerve into three branches; its anatomical photograph is shown below. Three applied techniques of surgical procedures: A—peroneal nerve graft to a tibial nerve without window incision, B—the same procedure with window incision, C—free graft of peroneal nerve using same nerve when its part was rotated 180°and sutured again.

**Figure 6 ijms-24-10428-f006:**
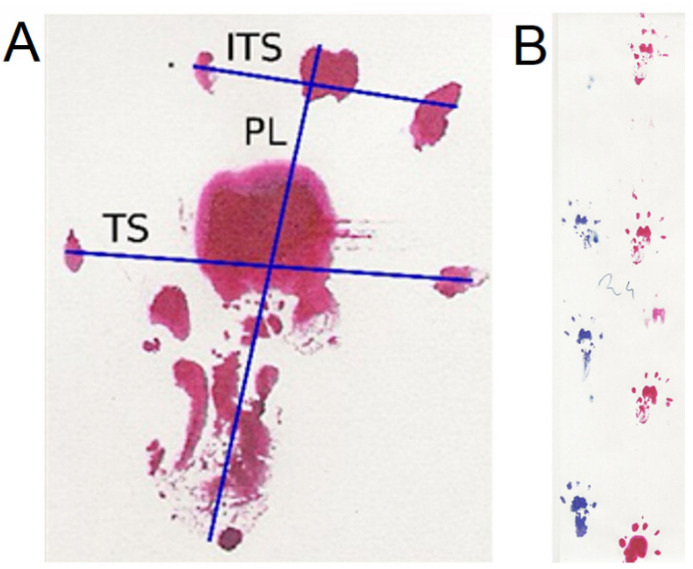
(**A**) Parameters measured in the footprint in the walking track analysis. (**B**) Example of the recording in walking track analysis of one animal from group A (end-to-side suture without an incision of the donor nerve epineurium) during locomotion. PL—print length, TS—toe spread, ITS—intermediate toe spread.

**Figure 7 ijms-24-10428-f007:**
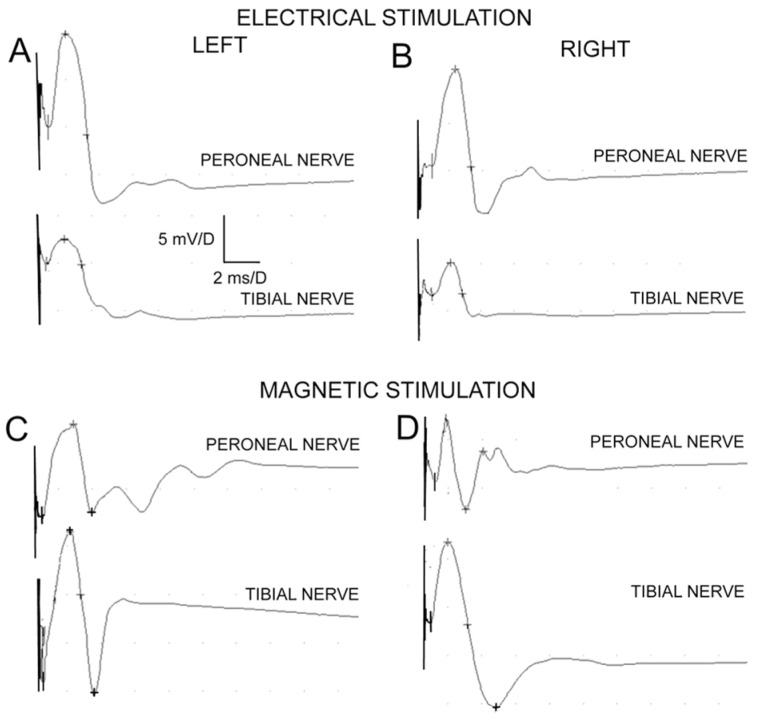
Examples of evoked potentials recorded from peroneal nerves and tibial nerves following electrical (**A**,**B**) and magnetic (**C**,**D**) stimulations in one of rats from experimental group A (end-to-side suture without an incision of the donor nerve epineurium). Calibration bars for amplification and time base presented in A are the same for all recordings. The first vertical cursor after the stimulus artifact indicates the latency onset of potential in ms for each recording. The strengths of applied stimulations are (**A**)—0.1 mA, (**B**)—0.1 mA, (**C**)—0.6 T, and (**D**)—0.7 T.

**Figure 8 ijms-24-10428-f008:**
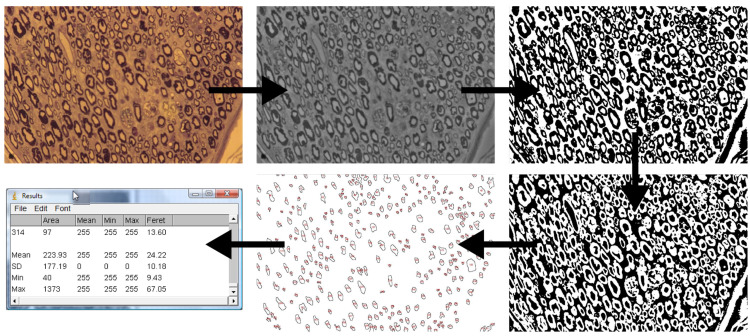
The algorithm of the microscopic photographs (40× image magnification) transformation being the histological transverse sections of the nerve aiming to evaluate the morphometric values (preliminary picture, conversion to 8-bit grayscale, threshold result, inversion, particle count, numerical results, respectively).

**Table 1 ijms-24-10428-t001:** Significance levels of differences between the PFI/TFI/SFI mean values at the end point of the study. Relationship between PFI, TFI, and SFI and follow-up time and between PFI and TFI in the studied groups (important relationships are marked bold).

		Significance Level LSD Test			Correlation between PFI, TFI, and SFI and Follow-Up Time		Correlation between PFI and TFI		
Tested Site		Group A	Group B	Group C	r	*p*-Value	r	*p*-Value	
Peroneal nerve (PFI)	**Group A** (N = 15)		0.66	0.67	**0.49**	**0.0001**	**0.74**	**<0.05**	Tibial nerve (TFI)
	**Group B** (N = 15)	0.66		0.38	**0.39**	**0.003**	**0.85**	**<0.05**	
	**Group C** (N = 15)	0.67	0.38		**0.23**	**0.0001**	**0.55**	**<0.05**	
Tibial nerve (TFI)	**Group A** (N = 15)		0.78	0.23	**0.43**	**0.0001**			
	**Group B** (N = 15)	0.78		0.33	**0.44**	**0.0001**			
	**Group C** (N = 15)	0.23	0.33		**0.31**	**0.0001**			
Sciatic nerve (SFI)	**Group A** (N = 15)		0.73	0.63	**0.43**	**0.0001**			
	**Group B** (N = 15)	0.73		0.89	**0.43**	**0.0001**			
	**Group C** (N = 15)	0.63	0.89		**0.28**	**0.0001**			

Abbreviations: PFI—peroneal function index; TFI—tibial function index; SFI—sciatic function index; r—rank correlation of the test results; *p*-values ≤ 0.05 are assumed rank correlation to indicate statistically significant differences.

**Table 2 ijms-24-10428-t002:** Comparison of the results from neurophysiological tests in three groups of rats (cumulative data). The mean values with standard deviations are presented. Significant values are marked in bold.

Type of Test Recording Site		Measured Parameter	Non-Operated Side Control			Operated Side			Non-Operated vs. Operated *p*-Value			ANOVA Variance Analysis between Operated Groups *p*-Value
			Group A (N = 14)	Group B (N = 12)	Group C (N = 14)	Group A (N = 14)	Group B (N = 12)	Group C (N = 14)	Group A	Group B	Group C	
Peroneal nerve	ENG	Amplitude (µV)	8000 ± 4459	10,818 ± 6856	7078 ± 5847	7043 ± 3521	6908 ± 2524	6414 ± 4311	0.54	0.08	0.73	0.89
		Latency (ms)	0.89 ± 0.20	0.88 ± 0.13	1.01 ± 0.38	1.02 ± 0.27	0.92 ± 0.27	0.96 ± 0.25	0.18	0.62	0.70	0.67
	MEP	Amplitude (µV)	9238 ± 6243	11,992 ± 7939	8013 ± 5614	7354 ± 4824	9958 ± 6325	10,514 ± 5540	0.40	0.50	0.24	0.31
		Latency (ms)	1.59 ± 0.36	1.57 ± 0.19	1.70 ± 0.29	1.56 ± 0.65	1.51 ± 0.32	1.57 ± 0.34	0.89	0.61	0.28	0.94
Tibial nerve	ENG	Amplitude (µV)	8639 ± 4318	12,845 ± 7257	9354 ± 5661	5657 ± 2913	5915 ± 4923	6141 ± 5199	**0.04**	**0.01**	0.25	0.96
		Latency (ms)	1.00 ± 0.31	0.92 ± 0.27	0.98 ± 0.40	1.14 ± 0.40	1.20 ± 0.49	1.04 ± 0.27	0.35	0.11	0.37	0.60
	MEP	Amplitude (µV)	9508 ± 6161	12,583 ± 6395	7408 ± 4645	6485 ± 4435	7133 ± 2927	7454 ± 3331	0.17	**0.01**	0.96	0.83
		Latency (ms)	1.71 ± 0.31	1.60 ± 0.40	1.66 ± 0.29	1.51 ± 0.35	1.65 ± 0.28	1.52 ± 0.22	0.14	0.75	0.15	0.40

Abbreviations: ENG—electroneurography; MEP—motor evoked potential; *p*-values ≤ 0.05 are assumed for comparison of mean values and for ANOVA variance analysis to indicate statistically significant difference.

**Table 3 ijms-24-10428-t003:** Mean and standard deviation of the analyzed morphometric parameters and *p*-values of Scheffe post-hoc ANOVA variance analysis (number, diameter, and surface area) of differences between groups and the norm for the peroneal and tibial nerves (significant values are marked bold).

Analyzed Morphometric Parameters			Number of Fibers					Diameter (um)					Surface Area (um^2^)				
			Variance Analysis between Groups					Variance Analysis between Groups					Variance Analysis between Groups				
Tested Site			Mean SD	Control	Group A	Group B	Group C	Mean SD	Control	Group A	Group B	Group C	Mean SD	Control	Group A	Group B	Group C
				(n = 15)	(n = 8)	(n = 9)	(n = 13)		(n = 15)	(n = 8)	(n = 9)	(n = 13)		(n = 15)	(n = 8)	(n = 9)	(n = 13)
Peroneal nerve	**Control**	353 ± 149			0.06	0.27	0.17	30 ± 7		**0.004**	**0.0004**	**0.003**	483 ± 234		**0.0004**	**0.0006**	**0.006**
	**Group A**	245 ± 119		0.06		0.45	0.5	22 ± 5	**0.004**		1	1	226 ± 123	**0.004**		1	1
	**Group B**	293 ± 138		0.27	0.45		0.88	21 ± 2	**0.0004**	1		1	199 ± 27	**0.0006**	1		1
	**Group C**	284 ± 103		0.17	0.5	0.88		22 ± 3	**0.003**	1	1		240 ± 69	**0.004**	1	1	
Tibial nerve	**Control**	399 ± 158			0.98	0.34	0.56	31 ± 8		0.51	0.15	0.84	477 ± 202		0.59	0.29	0.49
	**Group A**	397 ± 191		0.98		0.4	0.58	33 ± 5	0.51		0.47	0.72	518 ± 128	0.59		0.63	0.28
	**Group B**	329 ± 174		0.34	0.4		0.19	36 ± 10	0.15	0.47		0.31	559 ± 231	0.29	0.63		0.13
	**Group C**	446 ± 199		0.56	0.58	0.19		32 ± 8	0.84	0.72	0.31		418 ± 136	0.49	0.28	0.13	

## Data Availability

The datasets analyzed for this study can be found in the repository of the first author.

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
