# Peer review of "End-to-Side vs. Free Graft Nerve Reconstruction—Experimental Study on Rats"

_ijms, 2023, doi:10.3390/ijms241310428_

Round 1
Reviewer 1 Report
It is an interesting article dealing with peripheral nerve regeneration in which it is tested an end-to-side suture comparing it with the standard autologous graft. The experiment is well designed and the methodoloy appropriate using walking track, electroneurography and quantitative histology. Nonetheless, the paper does not provide new significant data, but supporting former papers. The introduction should include references about other experimental approaches on this topic besides artificial neurotube and grafted vein. In the introduction one can observe a contradiction by the authors since they think the comparisons between the end-to-side suture with the end-to-end suture is inadequate (lines 87-88), but they use an end-to-end suture as a control of the end-to-side experiment.
In line 113 the authors state that the donor was the tibial nerve, however, in group C the donor was the peroneal nerve since the fibers grew from the proximal stump of the same peroneal nerve.
Figure 2 caption should present an explanation of the acronyms appearing in A, and an example of the walking track of one animal from group B should be shown to illustrate the comparison with group A.
The explanation of tissue preparation (lines 269-274) is deficient. Please rewrite it and avoid "ossified", and mention the embedding steps.
The results about average number of fibers shown in figure 6 (and stated in line 583) are not consistent with what is shown in figure 7B in which the number of fibers is markedly lower in operated nerve. An automatic counting may lead to mistake actual nerve fibers with degenerating fibers devoid of axons, therefore a more detailed histological examination should be performed to rule out the question.
Photomicrographs shown in figure 7 have poor quality, showing shades and some are out of focus. Please replace the figures applying Köhler illumination and good focus. Also, it should be convenient to indicate the magnification and, in figure 7B caption, to which group the photos belong.
acceptable
Author Response
Answer attached into file

Reviewer 2 Report
The authors provide a manuscript on EtS nerve transfers with or without epineural window compared to autologous nerve transplantation. The manuscript is well written and technically sound. displayed methods and results are comprehensible. The outcomes provide an interesting influence for clinical application. As we all are aware of outcomes of rat trials are not directly translatable into human neuroregeneration I would encourage the authors to discuss their believed clinical application of their findings in the discussion section.
Author Response
Answer attached into file

Reviewer 3 Report
The article investigated the differences between a nerve side graft with and without an incision of the donor nerve epineurium. They used functional testing to assess recovery. They analyzed the walking tracks of animals, both toe spread and intermediate toe spread. Additionally, they did electrophysiologic assessments and morphometric analyses of the nerves. They found no differences between the groups suggesting an end-to-side nerve graft was just as good as traditional nerve grafts.
Recommendations:
- Please provide a clear hypothesis for the article in the introduction section. This addition will help future readers understand what you are investigating.
- Please include a scale bar in all figures with tiff or jpeg images.
- In Table 1, please fix the alignment of the groups. The term group is all on one line for the first column, while in the other columns, it is split between two lines. Additionally, why do you show both Scheffe and LSD tests? You should only show one. Make sure your discussion matches whichever test you use. Add to the method section which post-hoc test was used for Walking track analysis, histomorphometry, and electrophysiology testing.
- Figure legends for Figures 5 and 6 are the same, but different data is shown. Please be specific, as the data differs between the two figures, and future readers could easily be confused. Please include a title in the figure legend to indicate what results are being displayed.
- The images in Figure 7 are blurry, and the color appears off in a couple of them. Please try to get better representative images. Additionally, you may want to include arrows and stars indicating to future readers what you want them to notice. Do you have representative images for all the groups? Please include the groups not included and label them. It would be good to see the three groups side by side.
- In your discussion, you discuss the methods with great detail, making it unnecessarily long. Lines 431-458 appear unnecessary or should be significantly reduced. The statement from 480-481 that starts with "These results" should be expanded. Which results are similar, the end-to-side repairs are similar to previous studies? How do these results compare to studies that used an opening in the epineural sheath? I would also recommend adding these discussions to the other sections, i.e., 4.2 and 4.3. Where your results differ, please provide a possible explanation.
- In your summary/conclusion, state what this article adds to the field.
Edits:
- I recommend you spell out "XX" for the twentieth century in the abstract in line 17. It can be confusing how you have it written right now.
- Please review the paper to ensure easy readability. Many sections are unnecessarily wordy.
Author Response
Answer attached into file
